# The Impact of City Ranking on Industry Shifting: An Empirical Study

Xinyu Wang , Wensen Wu * and Haodong Li

School of Economics and Management, North China University of Technology, Beijing 100144, China; xinyu_wang@ncut.edu.cn (X.W.)

* Correspondence: 20102050126@mail.ncut.edu.cn

**Abstract:** This paper focuses on whether city honor competition has led to a sectoral shift. The research argues that cities' actions in pursuing honor have led to their changing from the manufacturing sector to the service sector. This paper attempts to construct a theory from city competition to sector shift. The research methods used are year-by-year propensity score matching and the difference-in-difference method. The results of the regressions prove that a city honor competition leads to a shift from the manufacturing sector to the service sector. The true value of this effect is approximately between 2.3274 and 3.0393, showing that the city honor competition promotes a sectoral shift. The trend of the city's economy towards the service sector is evident in the competition. The robustness test proves that the model satisfies the matching equilibrium assumption. The placebo test proves that other unobserved factors do not affect the policy. The heterogeneity test finds that the larger the city size, the stronger the effect of city honors on the sector.

**Keywords:** city honor competition; service sector; differences-in-differences; propensity score matching

## 1. Introduction

People often refer to different city rankings to help them choose where they want to work and live. To attract talent, this means that there is some competition between cities. This competition includes a city's livability and economic and cultural impact, among other criteria. When evaluating a city, people generally think of its industrial sector. Two examples are London's financial sector and California's IT industry. A city's ability to win the competition and achieve a high ranking as a livable city is closely related to its industry sector. Mature and environmentally friendly industry sectors provide cities with high levels of sustainable development and livability.

In 2003, China began to pay attention to the environmental pollution problems caused by its economic development. To achieve the goal of pollution control, the central government set up a competition between cities. This competition focuses on the selection of livable cities. These cities must comply with the competition's strict requirements to be awarded the honor. The rewards of this competition are political promotion and allocation of development resources. Many cities participate in the honor competition. Under strict environmental requirements, cities have had to abandon their polluting manufacturing sectors and seek to develop services. Thus, the change in the sector is evident. Many proponents believe that this leads to city sectoral transformation through competition and, ultimately, to sustainable economic development. However, opponents argue that the sectoral shift is a rule of economics and has nothing to do with the competition.

Is city honor competition leading to a sectoral shift from manufacturing to service? We are interested in the answer to this question. Public choice theory holds that any political decision is an economic act. In the political market, government as supplier chooses optimal policies based on cost–benefit analysis. Government is not like firms, which have

profit maximization as their goal. Rather, governments seek to increase their size. This can be viewed as an increase in power. Public choice theory indicates that the government is a monopoly provider of many services, and monopolies often lack efficiency. The solution to inefficiency is to introduce competition mechanisms. One type of competition is to increase competition among local governments. The central government has designed an honor competition for this purpose. Our study is concerned with the extent to which this competition can enable government to act beyond its own self-interest and thereby bring about a more livable life for its citizens. Therefore, we conduct a quasi-natural experiment on this competition policy.

Our research makes contributions to the current literature. First of all, this paper presents a unique perspective. City honor competitions may not initially be good policy for cities. This differs from the findings of many previous studies. At the time of the research, the process is susceptible to external validity and the threat of estimation bias. Second, we consider the robustness of the regressions. This study compares a common support sample with a weighted regression sample. The original model is improved by year-by-year propensity matching scores. Thus, the study addresses the problem of coefficient instability and sample self-matching. Therefore, the estimation results of this study regarding the competition are reliable.

The subsequent sections are organized in the following way. The next section is the literature review. Section 3 describes the policy background and hypotheses. Section 4 explains the data and methods used to find the answers. Section 5 performs the parallel trend test. The regression results and robustness and heterogeneity analyses will be discussed in Section 6. The last section concludes our study.

## 2. Literature Review

Many studies have attempted to use the PSM-DID model to analyze the impact of city honor policies [1]. Chen and Mao used the DID model to examine the impact of civilized city selection on city tourism [2]. However, it neglects the conduction of an in-depth analysis of the concurrent trends. Fan and Zhang used the same approach to study the impact of selection on resource-based cities [3]. However, the studies lack an analysis of the outcomes of different samples. Yang et al. paid attention to this problem in their study on green innovation [4]. Nevertheless, the study does not penetrate deeply into the model's problem of self-matching. Shi et al. analyzed the impact of civilized cities on green development based on administrative competition theory [5]. However, the study did not provide an insightful description of the influence mechanism of industrial structure. Liu et al. found a significant effect of civilized city selection on city energy efficiency [6], but the study overlooks a key concern: is industrial optimization the same for different cities? Peng et al. tried to narrow down the scope of the study to examine ecological evolution [7]. However, the study is limited to the ecological aspects of analyzing civilized cities. Fan et al. comprehensively considered the relationship between ecology, industry, and civilized city selection [8]. Studies such those by as Liu [9], Hu et al. [10], and Han et al. [11] have examined the impact of specific events on cities' carbon emissions. Nevertheless, past studies have not attempted to establish a framework interconnecting carbon emissions, city ecology, and industrial structure.

Each of these studies is about the analysis of city competitions. City competition is a broad concept. The most important aspect of a competition is the economic dimension. The competition on the economic level comes mainly from the technological revolution [12]. In particular, the new technologies of the Internet are constantly being applied to city life [13,14], but competition has also included cultural influence and political influence. Nevertheless, the indicator most frequently used to evaluate cities is still livability. Different studies have considered the sources and the effects of an improvement in city competitiveness.

Researchers have tried to find answers among cities. They consider innovative streets and city planning as ways to enhance city interaction and to drive competition. Streets

have the dual function of offering both commuting and social space. The ever-increasing exchange of people leads to competition for street space. Competition leads to the enhancement of different areas within the city. Finally, this achieves an increase in the competitiveness of the city [15]. Some studies have argued that the most important concern is the correlation between city competition and city economic sectors. With the help of quantitative analysis, researchers have presented evidence of the development of the financial sector. The size, openness, efficiency, and structure of the financial sector profoundly influence city competition in different regions [16]. Incentives from outside have also become a cause of competition in the city.

One view is that the most important external motivation is political incentives. The evaluation of a city is related to the future of its officials. Political incentives clearly reduce pollution, and this effect has a spatial spillover. This means that the effects of policies can spread throughout the region. However, the study also acknowledges that this is a short-term effect [17].

Technological competition is also an aspect of city competition. Industry 4.0 offers a good opportunity [18]. The spread of the Internet and the development of blockchain technology has led to the development of new manufacturing and advanced services [19,20]. These sectors have replaced the polluting manufacturing sector. They offer a better environment for city residents.

These studies note that competition is good for cities. Competition leads to the movement of urban factors of production and to a green and livable environment, but is this type of competition efficient? Are the government's actions justified? Some studies have presented evidence to answer these questions. Some researchers have analyzed city competition through the perspective of transportation construction. Based on their results, they argue that the Matthew Effect exists in city competition. The Matthew Effect describes the way in which stronger cities get stronger, and weaker cities get weaker. Transportation expansion leads to population loss, which leads to city shrinkage [21]. Studies of Beijing have shown similar concerns. The quality of Beijing's habitat is improving. However, the gap between the different regions within the city is growing significantly. Medical, economic, and transportation factors are all influencing the gap between cities [22,23].

### 3. Background and Theoretical Mechanism

*3.1. Policy Background*

The city honor competition is called National Civilized City. It is one of the methods used to evaluate the livability of cities in China. An honor city meets the 72 criteria in the requirements. The criteria for the selection will be shown in the Appendix A. Environment and pollution are undoubtedly the focus of the selection. Although the assessment is strict, a significant number of cities still participate. According to the 2020 City Competition selection, the number of participating cities has reached 263. The final number of cities that have received the honor is 133. This is an increase of 95 and 44 over the fifth selection. It is hard to ignore the government's actions in this honor competition, especially in the sectoral shift.

*3.2. Theoretical Hypothesis*

**Hypothesis 1:** *The city honor competition motivates governments to create sustainable economic structures through ecological criteria.*

Signaling theory suggests that city competition and political promotion are closely linked. For competitive incentives, Liu's research [9] finds that 70.59% of the Mayors and 66.67% of the Municipal Party Secretaries who received the honor are promoted in 5 years. More than half of the senior officials are promoted before their next reassignment after being honored. This will motivate the government to adjust the economic sector to follow environmental standards. Similar to other competition selections, it includes a review

mechanism. This mechanism tries to avoid irrational actions of the government in the short term. In an honor competition, companies will also judge the costs and benefits. Specifically, the government adopts command and control over energy-intensive and polluting industrial companies. The government requires them to reduce their emissions in the short term. They must "voluntarily" change their energy-consuming production patterns or invest more in pollution control [24]. These policies drive up the cost of their products. This ultimately puts them at a competitive disadvantage in the marketplace. At the same time, the government increases financial support for green companies. Most of these enterprises are in the service sector [25]. This lowers the pollution control and innovation cost of this type of enterprise.

On the other hand, the migration of residents is also a very important influence. Competition between cities can actually be considered as competition for talent. This means that innovation-intensive industrial sectors often require a livable urban environment, just as Silicon Valley does. This is one of the reasons why the government set stricter environmental standards. City honors are a signal to attract people to move to this city.

**Hypothesis 2:** *The city honor competition has an adverse selection in a short time.*

A city honor competition has an adverse selection in its early years. The study must show that the honor competition selection time is roughly a few months. This length of time does not exceed two years, even including the prep time. This means that the government will quickly increase pressure on the polluting manufacturing sector in the short term. Command and control measures can lead to a situation in which companies will internalize the cost of emissions. However, the cost of directly shutting down the large companies involved is high. This leads to companies being allowed by default, in order to make production adjustments without shutting down production [26]. The adjustment results in higher investment and output targets, and these, in turn, promote higher output in the traditional manufacturing sector. The government's desire to reduce the share of polluting manufacturing in the short term is defeated. It creates an adverse selection situation [27].

This is not out of the question. From the perspective of pollution control science, command-and-control instruments are usually not cost-effective, and they lack efficiency. This is mainly because the government lacks appropriate information on emissions and is unable to know the emission curves of all firms. This causes the sum of marginal control costs and marginal damage costs to deviate from the intersection of cost-effectiveness under the general method of allocating responsibility for emissions. The same adverse selection problem occurred in the U.S. chloralkali industry concerning technology diffusion. A high degree of environmental regulation appears to have gradually led to the use of greener ion membrane battery technology and to the development of the chloralkali industry. In reality, however, a large number of chloralkali producers ceased production in the short term [28]. In this study, the intensity of environmental regulation due to a city's honor competition is not as high; rather, it is an indirect policy impact, but it presents a similar problem.

These environmental costs are transferred to two groups, workers and consumers. This depends on the substitutability of the product. If the product is not substitutable, then consumers will be forced to accept the result of higher product prices. If the product has many substitutes, it is not a wise choice to raise the price. Dismissing workers to reduce costs is the main choice of the company. We tend to believe that both consumer and worker interests will be damaged.

**Hypothesis 3:** *The city honor competition will develop the service sector in the long run, reducing the share of the manufacturing sector and achieving sustainable economic development.*

County governments are often the primary force behind policy implementation. We believe that the impact of competition on county governments is the most direct. This

implies that a more direct impact can be observed by selecting the county government. This paper speculates that the negative impact is short-term. After the honor competition, this negative impact will gradually disappear. In the long term, the sectoral structure will undergo an autonomous reorganization and optimization. This implies a "U-shaped" rebound. However, county governments need to face the problem of policy independence. The autonomy of the government, to a large extent, affects the effectiveness of the policy. Therefore, this study chose Beijing. As the capital, the Beijing government's arrangements for sectoral development are mainly at the macro level. It has delegated many of its powers to county governments [29]. Their policy autonomy is generally considered to be the same as that of the municipal government. However, we must also acknowledge that measuring this autonomy is a challenge. City honor will ultimately promote sustainable economic development. The final analytical framework is shown in Figure 1.

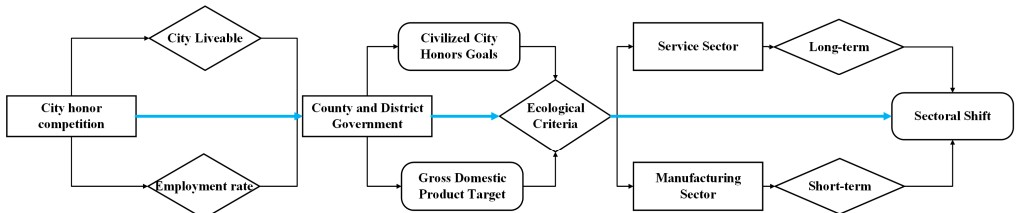

**Figure 1.** Theory Framework.

## 4. Study Design

### 4.1. Study Methodology

The propensity scores matching method with the differences-in-differences method (PSM-DID) is proposed to be used in this study. We choose carbon emissions, government expenditure, and social investment as control variables. These variables correspond to environment, government, and company. Furthermore, the study will face the problem of non-contemporaneous sample matching and endogeneity. The endogeneity lies in the fact that the 16 districts of Beijing are non-random samples. Regional development direction, economic development level, geographical environment differences, and other regional-specific factors may influence the policy's influence. Panel data are employed to construct a difference-in-difference model that assumes the findings of prior studies and that may be optimized by matching. It can solve the problem of non-contemporaneous sample matching and endogeneity [30].

### 4.2. Timing and Treatment Groups

To test the impact of the honor competition, this paper uses the civilized districts in Beijing as the treatment groups. Among them, Chaoyang District, Dongcheng District, Haidian District, Tongzhou District, Xicheng District, and Yanqing District are the treatment groups. The rest of the districts are the control group. The net effect of an impact is determined by comparing the treatment and the control groups. Each district implements the livable city policy at different points in time. In Beijing, most districts will start preparing for the competition two years in advance. Special mention needs to be made of the Tongzhou and Dongcheng districts. Much of their preparation is focused on the year before the competition. Therefore, we made the judgment that the other treatment groups are two years ahead of schedule, while the Tongzhou District and the Dongcheng District are only one year ahead of schedule. For the choice of timing, this study emphasizes the implementation of the honor competition policy rather than the timing of receiving the honor. It is just like an athlete participating in a competition. We cannot deny the improvement in ability that results from training just because the athlete failed to win a medal.

Many previous studies have a habit of lagging policies by one year. This may be a serious estimation bias and an exogenous threat to the city honor competition. The estimation bias stems mainly from the small estimated coefficients due to ignoring the pre-honor

competition policy effects. The exogenous threat is that the time point chosen for the study does not apply to other similar analyses.

*4.3. Econometric Model*

This paper constructs the PSM-DID model with multiple time points. The propensity score matching method considers other variables for individuals with the same or similar propensity values to have the same characteristics in the distribution. We matched the individuals in the treatment and control groups according to propensity values. Thus, the baseline data of the two sample groups are balanced. This method can achieve an effect similar to a random grouping. The difference-in-difference method is similar in principle to a natural experiment. It treats the implementation of a policy as a natural experiment. We set up a control group unaffected by the policy. The study can compare the analysis of the control group with that of the treatment group affected by the policy. This can obtain the net effect of the policy implementation on the study subjects.

The dependent variable in this paper is the sectoral shift (Sector). It represents the change in economic development from the manufacturing sector to the service sector. This paper draws on the treatment group of Gan et al. [31]. We choose to represent it as the ratio of value added in the manufacturing sector to value added in the service sector. The study sets the dummy variable Honor to analyze the effect of the policy. Honor is specifically the product of the selected city dummy variable treatment and the selected time dummy variable post.

The model is set as follows:

$$Honor_{it} = treated_{it} * post_{it} \tag{1}$$

$$Sector_{it} = \alpha_0 + \alpha_1 Honor_{it} + \alpha_2 xlist_{it} + \delta_{it} + \mu_{it} + \varepsilon_{it} \tag{2}$$

In Equation (1), $treated_{it}$ is used as a dummy variable of treatment groups. $post_{it}$ is a time dummy variable which is set to 1 after the district is selected as a civilized city and 0 before that. So, $DID_{it}$ is the interaction term of $treated_{it}$ and $post_{it}$. $Honor_{it}$ is used as the policy variable for each district civilization city selection.

In Equation (2), $Scetor_{it}$ represents the industrial structure optimization of 16 districts in Beijing. $Honor_{it}$ is the treatment variable. $xlist_{it}$ is other control variables. $\mu_{it}$ and $\delta_{it}$ denote city and time-fixed effects, respectively. $\varepsilon_{it}$ is the random variable. $i$ denotes the region, and $t$ denotes the year.

*4.4. Data Source and Variables*

Based on data availability, this paper selects the panel data of 16 districts in Beijing from 2003 to 2020. The data in this paper are obtained from the Beijing Regional Statistical Yearbook, the China Regional Statistical Yearbook, the China County Statistical Yearbook, and CEADs (the China Carbon Accounting Database) from previous years.

The study considers the robustness and the scientific validity of the results [31,32]. Environment, material capital, and government expenditure are selected as the control variables. Among them, carbon dioxide emissions (Enviro) is the environmental variable. Capital investment (Cap) is measured by the ratio of company fixed investment to GDP in each county. Government expenditure (Gov) is expressed through the share of fiscal expenditure in GDP. All variables, including GDP, are in million ¥. Table 1 shows the description of the variables.

**Table 1.** Descriptive Statistics.

| Name | Symbol | Unit | Average | Std. Dev | Min | Max |
|---|---|---|---|---|---|---|
| Sectoral shift | Sector | % | 4.108 | 6.044 | 0.404 | 39.486 |
| Civilized City Honor | Honor | (0.1) | 0.211 | 0.409 | 0.000 | 1.000 |
| Carbon dioxide emissions | Enviro | % | 4.062 | 3.470 | 0.601 | 16.413 |
| Capital investment | Cap | % | 0.637 | 0.402 | 0.637 | 2.180 |
| Government expenditure | Gov | % | 0.281 | 0.199 | 0.281 | 0.948 |

## 5. Parallel Trend Test and Analysis

### 5.1. Parallel Trend Test

The study is concerned with signaling results with the average meaning of the city honor in the megacity. However, this requires that the parallel trend assumption is satisfied. That is, the treatment and control groups must have the same trend before implementing the policy. If this condition is not satisfied, then the derived is not precisely the true policy effect.

The Figure 2 shows that the regression coefficients do not pass the test of significance level before the honor competition. This indicates that there is no significant difference between the treatment and control groups before the competition. At the time of city honors, the effect began to show up. This satisfies the requirement of the parallel trend. In further analysis, the coefficient of Sector is negative in the year in which it is named the civilized city. The two tests suggest that this negative effect will persist. Moreover, the coefficient gradually returns to a positive value over time.

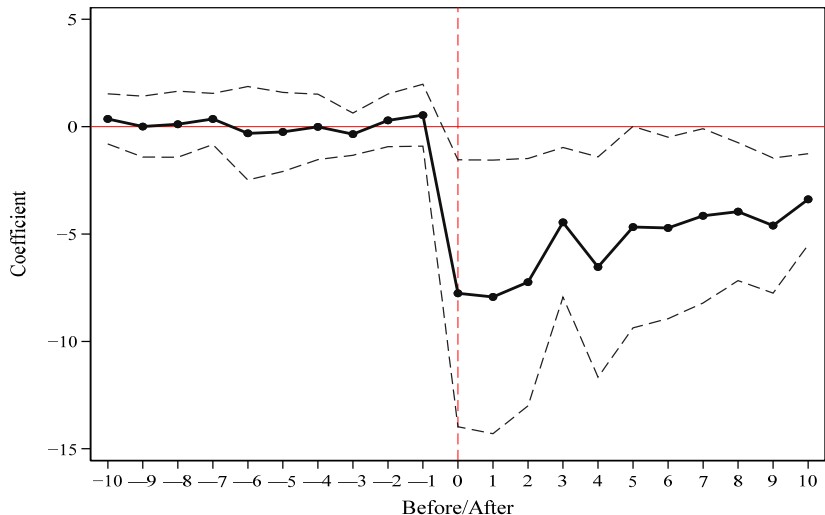

**Figure 2.** Parallel trend test.

### 5.2. Trend Analysis

Hypothesis 2 argues that governments are the most critical subjects in declaring honorable competitions. They often face the political test [33]. Both the honor of a livable city and the employment rate are what the government must pursue. This is also reflected in the actions of the government. Specifically, the government publishes various restrictive orders on polluting manufacturing sectors to meet the criteria of city honors. However, this measure is not without its costs. Higher costs have led companies to lay off workers or to reduce wages. This has led to a decrease in workers' spending power.

It is a painful choice. Generally, governments require manufacturing sectors to expand their investments in new production equipment and in clean technology. In exchange, the government allows them to continue production. At the same time, governments need to seek compensation for lost GDP in other ways. They create incentives to develop low-energy, high-value-added service sectors. As for those companies that accept

the government's request, investing in new equipment implies optimistic expectations for the long-term production of these companies. The expansion of production in the manufacturing sector in the short term is a result of increased investment. Investment in the manufacturing sector causes adverse selection in the short term.

In summary, this is the reason for the negative impact when it comes to starting the honor competition, and it takes some time for the sector to reorganize, so we can see that the results of these two tests show that the negative effect is gradually becoming smaller. Optimistically, there are reasons to believe in the U-shaped rebound in Hypothesis 3: the manufacturing sector is gradually being replaced by the service sector, and the structure of the economy is turning towards sustainable development.

## 6. Result Analysis

### 6.1. Benchmark Model

This study decides to select different samples to test the impact effect. These samples are of two types: those that meet common support (On_Support), and those that undergo frequency-weighted regression (Weight_Reg). The samples with common support meet the common support hypothesis. This hypothesis requires some overlap in the distribution of characteristics of the treatment and the control group samples to ensure matching quality. The samples after frequency-weighted regression indicate that the matched samples can be matched as multiple treatment groups. The treatment group samples were replicated according to weight size. The results are shown in Table 2.

**Table 2.** Results of the differences-in-differences model.

|  | (1) | (2) | (3) | (4) | (5) | (6) |
|---|---|---|---|---|---|---|
| Honor | 10.5127 *** | 8.7223 *** | 3.4655 *** | 1.7923 *** | 1.7923 ** | 1.4305 * |
|  | (9.8559) | (9.8352) | (7.1814) | (3.8637) | (2.4149) | (1.9364) |
| Enviro |  | −0.3624 *** | 0.0209 | −0.5858 *** | −0.5858 ** | −0.6706 * |
|  |  | (−4.5663) | (0.1551) | (−3.4981) | (−2.3069) | (−1.8699) |
| Cap |  | −2.1930 *** | −2.4593 *** | −0.7092 | −0.7092 | −0.8294 |
|  |  | (−4.8342) | (−3.9159) | (−1.1979) | (−0.9245) | (−0.9346) |
| Gov |  | −4.8624 *** | 0.9011 | −4.7086 *** | −4.7086 * | −5.5630 * |
|  |  | (−4.4290) | (0.6645) | (−2.9739) | (−1.9973) | (−1.9181) |
| District Fixed | No | No | Yes | Yes | Yes | Yes |
| Time Fixed | No | No | No | Yes | Yes | Yes |
| Cluster | No | No | No | No | Yes | Yes |
| On_Support | No | No | Yes | Yes | Yes | No |
| Weight_Reg | No | No | No | No | No | Yes |
| N | 288 | 288 | 277 | 277 | 277 | 281 |
| r2_a | 0.5050 | 0.6043 | 0.8966 | 0.9216 | 0.9213 | 0.9229 |

Note: T-value are in parentheses, *, **, and *** indicate significance at the 10%, 5%, and 1% levels, respectively.

In five of the six regressions, the results are significant at the honor 5% level. The regression in Column 2 adds omitted control variables to the regression in Column 1. The coefficient of honor decreases from 10.5127 to 8.7223. The regression in Column 4 uses a differences-in-differences model and adds time-fixed effects and individual-fixed effects. This results in an estimated coefficient of honor of 1.7923, while the R-squared is significantly increased. Column 5 adds clustering robust standard errors. The significance of the regressions remains significant at the 5% level. This proves that the city honor race can drive the service sector to replace the manufacturing sector.

The results in the control variables show that reducing carbon emissions can also drive the services sector to displace the manufacturing sector. Social investment is not significant in the last three regressions. The coefficient of government expenditure is significantly negative. This value is large. This proves the important influence of the government on the sectoral shift. It also supports our hypothesis that the government does not act rationally in honor competitions.

However, we still must question whether this result is biased. The cross-period matching of the sample may have led to an underestimation of the estimated coefficient values. This judgment comes mainly from the fact that in the regression in Column 6, the coefficient of the weighted regression is smaller than the coefficient of the common support. Its coefficient is also not significant at the 5% level.

### 6.2. Year-by-Year Propensity Score Matching

Referring to the methods of Heyman et al. [34] and Böckerman and Ilmakunnas [35], the study uses year-by-year propensity score matching. After the samples are matched, it uses the differences-in-differences model. Specifically, the study first performs the propensity score matching method for each year of the sample. We then collect and integrate the propensity score results for each year for regression. In short, this method focuses on matching for each year. The sample is matched mainly in the current year. This does not result in matching the 2003 sample with the 2020 sample. Finally, the study conducts a two-way fixed effects regression. The regression results are shown in Table 3.

**Table 3.** After year-by-year propensity score matching.

|  | (1) | (2) | (3) | (4) | (5) | (6) |
|---|---|---|---|---|---|---|
| Honor | 10.4143 *** | 8.7211 *** | 5.3639 *** | 2.3274 ** | 2.3274 ** | 3.0393 *** |
|  | (9.8899) | (9.8934) | (4.5655) | (2.0240) | (2.3371) | (3.4118) |
| Enviro |  | −0.3532 *** | −0.1054 | −1.1599 *** | −1.1599 ** | −1.1060 ** |
|  |  | (−4.5566) | (−0.3577) | (−3.7053) | (−2.8692) | (−2.7901) |
| Cap |  | −2.2140 *** | −7.7182 *** | −4.3438 ** | −4.3438 * | −5.0472 ** |
|  |  | (−4.9025) | (−3.1221) | (−2.1558) | (−1.8680) | (−2.3866) |
| Gov |  | −4.6274 *** | 9.8622 | −8.9207 | −8.9207 | −8.4789 |
|  |  | (−4.3379) | (1.4585) | (−1.2033) | (−1.1792) | (−1.0703) |
| District Fixed | No | No | Yes | Yes | Yes | Yes |
| Time Fixed | No | No | No | Yes | Yes | Yes |
| Cluster | No | No | No | No | Yes | Yes |
| On_Support | No | No | Yes | Yes | Yes | No |
| Weight_Reg | No | No | No | No | No | Yes |
| r2_a | 0.4971 | 0.5941 | 0.8572 | 0.9144 | 0.9127 | 0.9230 |

Note: T-value are in parentheses, *, **, and *** indicate significance at the 10%, 5%, and 1% levels, respectively.

We are mainly interested in the regression results for Columns 3–6. It is found that the coefficients and the significance of honor are higher than the results of the benchmark regression. This supports our judgment that the traditional method would underestimate the true estimates. In particular, the results of Regression 5 and Regression 6 are compared. The coefficient of Column 6 is 3.0393, which is higher than the coefficient of Column 5 at 2.3274. This reflects the improved weighted coefficient. We have reason to believe that the true value is between 2.3274 and 3.0393. This evidence represents that the honor competition has more impact than imagined.

Meanwhile, the carbon emission coefficient is also significantly negative in most regressions. The coefficient for suppressing carbon emissions is also underestimated when comparing the benchmark regressions. All of this evidence suggests that year-by-year propensity score matching is valid. There is a slight improvement in the adjusted R-squared.

### 6.3. Robustness Test

6.3.1. Propensity Matching

It needs to be confirmed whether the matching equilibrium hypothesis is satisfied. The main component of the hypothesis is whether there is a significant difference between groups before and after matching. Table 4 shows that the *p*-values of the matched covariates become insignificant, which means that there is no significant difference between the treatment group and the control group. Meanwhile, the standard deviation of all covari-

ates decreased substantially. Except for the deviation of government intervention higher than 10%, the deviation of other covariates is below 10%.

**Table 4.** Comparison results of propensity matching.

| Variables | Before Matching | | | After Matching | | |
|---|---|---|---|---|---|---|
| | Bias (%) | t | *p* Value | Bias (%) | t | *p* Value |
| Enviro | 29.60 | 2.65 | 0.01 | −0.39 | −0.06 | 0.69 |
| Cap | −74.40 | −6.16 | 0.00 | −0.63 | −0.37 | 0.53 |
| Gov | −43.60 | −3.68 | 0.00 | −15.6 | −1.14 | 0.26 |

### 6.3.2. Kernel Density Test

The study is further estimated using the kernel matching method. The robustness of the effect of DID is tested. The density function plots of the propensity score values are shown in Figures 3 and 4. The probability densities of the propensity scores are closer after matching, both for direct propensity score matching and for year-by-year propensity score matching. The shortening line distance between the treatment and control groups supports these findings. Moreover, the year-by-year propensity score matching is more effective. Therefore, using the year-by-year propensity score matching is a better method.

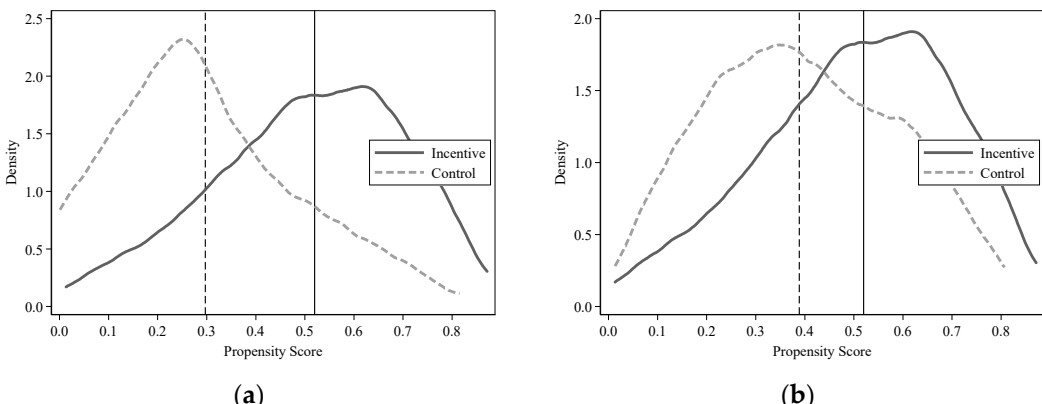

**Figure 3.** Kernel density test: (**a**) before direct matching (**b**) after direct matching.

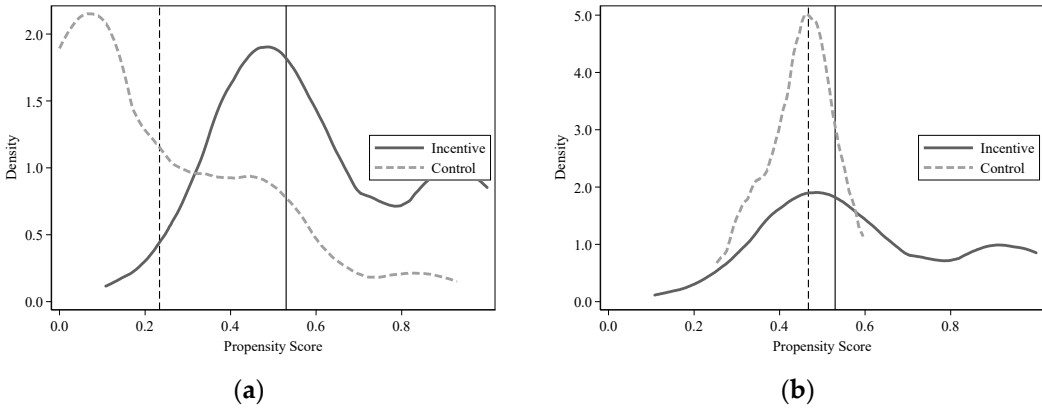

**Figure 4.** Kernel density test: (**a**) before year-by-year matching (**b**) after year-by-year matching.

### 6.3.3. Placebo Test

This study refers to the existing literature for a placebo test to disprove the notion that unobserved factors influence DID [36,37]. In this study, a random sampling process is applied to determine the treatment and the control groups. This randomization eliminates selection bias. The randomized areas are then combined into the original data set that has

been processed. Iterative regressions of the randomized interaction terms are performed to obtain the estimated coefficients 1000 times. The results are displayed in Figure 5.

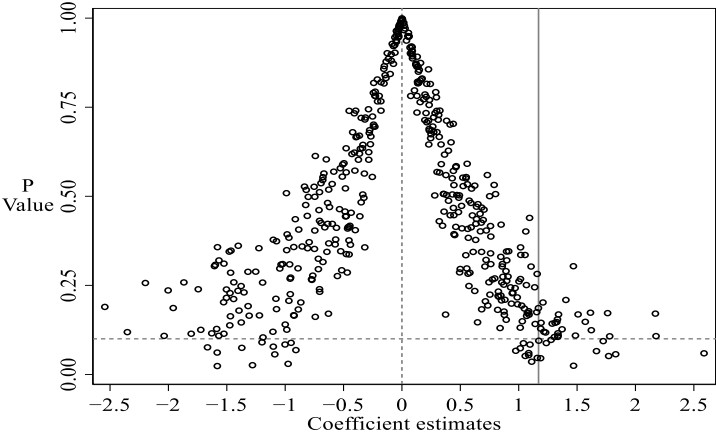

**Figure 5.** Placebo test results.

The test results show that most of the coefficients are concentrated around 0. Moreover, their coefficient averages are far from the actual values of the coefficients. Most of the scatter points are located above the dashed line. This indicates that most of the estimated coefficients are insignificant at the 10% level. This also implies that other unobserved factors do not influence the effect of the city honor competition.

*6.4. Heterogeneity Analysis*

6.4.1. District Functions

Since the reform and the opening up, China has implemented an unbalanced development model that prioritizes efficiency. This development model has led to the dominant role of central districts in development. This has led to a widening gap between districts. The central districts have created a strong siphon effect [38]. If the degree of industrial isomorphism between districts is higher, the central districts' siphoning effect is stronger. Therefore, for districts with different functions, this paper considers the impact of honor competitions to be different.

The functions of each district are determined by the Beijing Main Functional Area Plan. The sample is divided into two groups; the first group (Centr) is the group of central districts. The second group (Frin) is the group of fringe and satellite districts. The study does not directly follow the four categories in the plan because, first, the districts in the first group are generally located within the central ring of Beijing. In contrast, the districts in the second group are distributed around Beijing. Categorizing them can better fit the concept of central and peripheral city areas. Second, the sample of cities in the partial classification needs to be bigger. The direct use of fixed effects tends to cause the coefficients of the interaction terms to be ignored.

Table 5 shows that the coefficient of honor for the central districts is 3.0396. The coefficient of the fringe city districts is −1.2912. Table 4 shows that the central districts have a strong desire to build a livable city. High-value-added and more environmentally friendly service sectors are preferred in these districts. In addition, they have abundant resources to allocate. Taking advantage of the honor city competition, they can mobilize richer resources to change the proportion of economic sectors. In the honor competition, fringe districts take over the transfer of manufacturing sectors from the central districts. To avoid a high degree of sectoral isomorphism with the central districts, they are not keen to imitate the development model of the central districts.

**Table 5.** Heterogeneity analysis of district functions.

|  | *Group A* | *Group B* |
|---|---|---|
| Sector_Centr | 3.0396 *** | |
| | (4.0195) | |
| Sector_Frin | | −1.2912 * |
| | | (−1.8046) |
| Control variables | Yes | Yes |
| District Fixed | Yes | Yes |
| On Support | Yes | Yes |
| Time Fixed | Yes | Yes |
| Cluster | Yes | Yes |
| N | 277 | 277 |
| r2_a | 0.9263 | 0.9173 |

Note: T-value are in parentheses, *, and *** indicate significance at the 10%, and 1% levels, respectively. The samples in Group A are the central city districts and in Group B are the fringe city districts.

## 6.4.2. District GDP

Different GDP sizes drive districts to choose to develop different sectors. Considering the sample size limitations, the division dummy variables are only set internally. The study uses the GDP maximum at the sample time as the benchmark. The GDP percentage of each district at different points in time is used as the basis for dividing the dummy variables. This division uses 20% as the division interval. However, the sample size of the boundary is small. If fixed effects are applied to the sample of boundary intervals, they can easily cause the interaction term coefficients to be ignored. Also, to maintain the balance of the sample size in each interval, cities reaching 100–60% are classified as large districts (Big), cities reaching 60–40% are classified as medium districts (Middle), and cities reaching 40% and below are classified as small districts (Small). The interaction term variables are constructed on this basis (shown in Table 6).

**Table 6.** Heterogeneity analysis of district functions.

|  | *Small Districts* | *Middle Districts* | *Big Districts* |
|---|---|---|---|
| Sector_Small | −0.2820 | | |
| | (−0.2295) | | |
| Sector_Middle | | 1.7916 ** | |
| | | (2.3186) | |
| Sector_Big | | | 2.6038 ** |
| | | | (2.6331) |
| Control variables | Yes | Yes | Yes |
| District Fixed | Yes | Yes | Yes |
| On Support | Yes | Yes | Yes |
| Time Fixed | Yes | Yes | Yes |
| Cluster | Yes | Yes | Yes |
| N | 277 | 277 | 277 |
| r2_a | 0.9165 | 0.9196 | 0.9200 |

Note: T-value are in parentheses, ** indicate significance at the 5% levels, respectively. Small districts are the samples with GDP ranking in the last 40%; middle districts are the samples with GDP ranking in the 40–60%; and big districts are the samples with GDP ranking in the top 40%.

The regression results in Table 6 show that the coefficient of DID is significantly positive for medium and large city districts. The coefficient for large city districts is 2.6038, which is higher than that of 1.7916 for medium districts. The coefficient for small city districts is negative. Also, the coefficient is insignificant. This represents the need for larger cities to have more capacity and willingness to participate in this policy. Smaller cities lack the willingness and ability to do so. This result again echoes the results of the previous heterogeneity analysis. It suggests that the larger the city, the greater the effect of city honors on the shift from the manufacturing sector to the service sector.

## 7. Conclusions and Limitations

This paper focuses on whether city honor competitions have led to sectoral shifts. The paper attempts to construct a theory of city competition to the sectoral shift. The results of the regressions demonstrate that the city honor competition leads to a change from the manufacturing sector to the service sector. The year-by-year matching method can solve the problem of low estimated coefficients. The true coefficient of city honors in economic sectors is approximately between 2.3274 and 3.0393. That is, city honor competition promotes a sectoral shift. In the competition, the trend of the city economy towards the service sector is evident. The robustness test proves that the model satisfies the matching equilibrium assumption. The placebo test proves that other unobserved factors do not influence the policy. The heterogeneity test finds that the larger the city size, the more the city honor can promote the replacement of the manufacturing sector by the service sector.

This study has some policy implications. It suggests that competition designers should be aware of the irrationality of government action and the heterogeneity among cities. We aim to remind the competition's designers of this. However, we have to acknowledge the limitations of this study. First, the political aspect is missing from the analysis. The relationship between this honor competition and politics is obvious. The limitations of publicly available data make it difficult for us to analyze more explicit political associations. Second is the problem of omitted variables. Shifting in economic sectors can be influenced by many factors. Human capital, globalization, and technological progress can all be seen as control variables. These are points that could be further studied in the future.

**Author Contributions:** Conceptualization, W.W. and X.W.; methodology, W.W. and X.W.; software, W.W.; validation, W.W.; formal analysis, W.W.; resources, X.W.; data curation, W.W. and H.L.; writing—original draft preparation, W.W. and H.L.; writing—review and editing, W.W.; visualization, W.W.; supervision, X.W.; project administration, X.W.; funding acquisition, X.W. All authors have read and agreed to the published version of the manuscript.

**Funding:** The article is supported by the Beijing Social Science Foundation Youth Project (no. 21JJC021), Social Science Program of Beijing Municipal Education Commission (no.SM202210009007), and the 2023 Beijing Innovation and Entrepreneurship Training program for University Students.

**Institutional Review Board Statement:** Not applicable.

**Informed Consent Statement:** The story does not need ethical approval.

**Data Availability Statement:** The datasets used and/or analyzed during the current study are available from the corresponding author upon reasonable request.

**Conflicts of Interest:** The authors declare no conflict of interest.

## Appendix A

City Ranking Criteria

The city honor competition is also called National Civilized City. There are 72 criteria for the civilized city in Table A1. These criteria relate to politics, economy, culture, environment, urban-rural equity, ethnic equity, and government administration. For better understanding, we paraphrase the criteria. Table 1 shows the criteria with which cities in the competition need to comply. The competition is based on a point deduction system out of 100, i.e., points will be deducted if the criteria are not met. If the criteria are seriously violated, the city will be disqualified.

**Table A1.** City honor competition criteria.

| No. | Criteria |
| --- | --- |
| 1 | Party Leader Education |
| 2 | Political Propaganda |
| 3 | National Spirit Propaganda |
| 4 | Ideological Propaganda |
| 5 | Infrastructure for the Media |
| 6 | Citizen Moral Cultivation |
| 7 | Citizen Values Cultivation |
| 8 | Traditional Cultural Heritage |
| 9 | Green Lifestyle Popularization |
| 10 | Citizen Honesty Cultivation |
| 11 | Citizen Travel |
| 12 | Citizen Traffic |
| 13 | Citizen Network |
| 14 | Praise for Role Models |
| 15 | Public Service Announcements |
| 16 | Citizen Health |
| 17 | Basic Government Work |
| 18 | Public Service Provision |
| 19 | Charity Service |
| 20 | Community Welfare Services |
| 21 | Future Development Plan |
| 22 | Rural Development |
| 23 | Individual Social Responsibility |
| 24 | Family Social Responsibility |
| 25 | School Social Responsibility |
| 26 | Party Integrity |
| 27 | Simplify Government Administration |
| 28 | Disclosure Government Affairs |
| 29 | Government Democratization |
| 30 | Community Democratization |
| 31 | Civil Rights Protection |
| 32 | Safeguarding Vulnerable People |
| 33 | Social Credit |
| 34 | Penalties for Defaulters |
| 35 | Government Service Improvement |
| 36 | National Compulsory Education |
| 37 | Management Education Company |
| 38 | Science Popularization |
| 39 | Cultural Investment |
| 40 | Cultural Product Supply |
| 41 | Cultural Infrastructure |
| 42 | Cultural Industry |
| 43 | Ethnic Equality and Unity |
| 44 | Increase Productivity |
| 45 | Raising Citizens' Income |
| 46 | Smart City Construction |
| 47 | Barrier Free Facilities |
| 48 | Excellent Urban Environment |
| 49 | Excellent Urban Cleanliness |
| 50 | Excellent Urban Order |
| 51 | Harmonious Community |
| 52 | National Health Protection |
| 53 | Army and People in Harmony |
| 54 | Public Safety |
| 55 | Drug Safety |

**Table A1.** *Cont.*

| No. | Criteria |
| --- | --- |
| 56 | Production Safety |
| 57 | Punishing Crime |
| 58 | Preventing Public Crisis |
| 59 | Urban Air Quality |
| 60 | Urban Water Quality |
| 61 | Clean Urban Drinking Water |
| 62 | Sustainable Urban Economy |
| 63 | Urban Noise |
| 64 | Environmental Concept Advocacy |
| 65 | Land Administration |
| 66 | Rural Industry |
| 67 | Rural Infrastructure |
| 68 | Urban and Rural Recourse Equity |
| 69 | Urban and Rural Public Services Equity |
| 70 | Citizen Participation in Management |
| 71 | Political Performance Assessment |
| 72 | Emergency Management |

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
