# Peer review of "The Impact of City Ranking on Industry Shifting: An Empirical Study"

_sustainability, doi:10.3390/su15118930_

Round 1

Reviewer 1 Report

This article examines the impact of city honor competition on Industry Shifting. The PSM and DID method are used as the estimation technique. The topic is undoubtedly important. The research has marginal potentials for practical implications. However, some aspects should be addressed in order to improve the overall quality of the paper:

1. The introduction is not well written. the introduction part is lack of theoretical depth. Moreover, the objective of this study is not well introduced in the first section. Please modify it as suggested.

 2. The structure of literature is also strange. Pls refer to the latest paper in Sustainability journal and revise the whole paper.

 3.Please refer to some references to support your variables selection

4. the design of the empirical analysis is wrong. Why do you select the R-sector as another dependent variable to conduct Parallel trend test. Not use your explained variables in the empirical analysis. In the Figure 2, I shows the parallel test failed to pass after 2 periods. please explain it.

5. what is the difference between OLS and DID. OLS also can used to conduct DID by generating dummy variables. The classification is wrong in Table 2.

 6. Robustness analysis can be enhanced

7.What are the policy implications of this study?

The quality of English language needs to be polished.

Author Response

Response Letter

We are thankful for the reviewers and editor’s comments. The manuscript is revised according to the reviewer’s advice. All changes were highlighted in RED in the manuscript.

Reviewer 1:

  1. The introduction is not well written. the introduction part is lack of theoretical depth. Moreover, the objective of this study is not well introduced in the first section. Please modify it as suggested.

Response: Thank you for your valuable comment.

We add public choice theory to the introduction (line 42-52). Public choice theory holds that any political decision is an economic action. In the political market, government as supplier chooses optimal policies based on cost benefit analysis. It is not like firms that have profit maximization as their goals. The government seeks to increase its size. This can be viewed as an increase in power. Public choice theory indicates that the government is a monopoly provider of many services. And monopoly often lacks efficiency. The solution to inefficiency is to introduce competition mechanisms among local governments. The central government has designed an honor competition for this purpose. Our study is concerned with the extent to which this competition can enable government to act beyond its own self-interest and thereby bring about a more sustainable industrial structure.

  1. The structure of literature is also strange. Pls refer to the latest paper in Sustainability journal and revise the whole paper.

Response: Thank you for your comment.

Our article refers to the structure of Chen. Its methodology and themes are similar to our study. However, we do not have heading level 3. This could cause some misunderstanding, for example in the robustness test section. Therefore, we added heading level 3 to better show the relationship between the different sections. (line 384-385、394、416、431-432、461)

Chen, Q.; Mao, Y., Do City Honors Increase Tourism Economic Growth? A Quasi-Natural Experimental Research Study Based on “Civilized City” Selection in China. Sustainability 2021, 13, (22), 12545.

  1. Please refer to some references to support your variables selection

Response: Thank you for your comment. We have added the corresponding references about variables selection. (line 281)

Gan, C.; Zheng, R.; Yu, D., An empirical study on the effects of industrial structure on economic growth and fluctuations in China. Economic Research Journal 2011, 5, 4-16.

Liu, Z.; Liu, C., Research on the effect of civilized city on the upgrading of industrial structure: A quasi-natural experiment from the selection of civilized city. Ind. Econ. Res 2021, 1, 43-55.

  1. the design of the empirical analysis is wrong. Why do you select the R-sector as another dependent variable to conduct Parallel trend test. Not use your explained variables in the empirical analysis. In the Figure 2, I shows the parallel test failed to pass after 2 periods. please explain it.

Response: Thank you for your valuable comment.

We set up a parallel trend test for two variables intended to show the test is robust. However, we seem to have overlooked a problem that is confused. We have corrected this and extended the test time. We find that after this adjustment, the results after the competition policy are significant. (line 296-304)

We provide further explanations. The rotation time of Chinese officials is about 5 years. This means that if a senior official is honored, he/she is likely to have been promoted in 5 years. According to Liu’s research, it finds that 70.59% of the Mayors and 66.67% of the Municipal Party Secretaries who received the honor are promoted in 5 years. More than half of the senior officials are promoted before their next reassignment after being honored. (line 142-145)

This phenomenon is important. The will of senior officials can represent the government in many cases. They have a strong incentive to compete and maintain the honor. But their successors do not have that motivation. The successors only need to prevent the worst case happening. That is, their city is stripped of its honor. They prefer to implement their own policies to show their contributions. So, it's a shock to the causal effect.

Liu, S. (2019). The Incentive of “Commendation” in Policy Implementation—Taking the Establishment of A Civilized City in China as An Example. Chin. Public Adm, 2, 72-78.

  1. What is the difference between OLS and DID. OLS also can used to conduct DID by generating dummy variables. The classification is wrong in Table 2.

Response: Thank you for your comment.

We have noticed the errors and corrected them, along to Table 3. (line 358、382)

  1. Robustness analysis can be enhanced

Response: Thank you for your comment.

To better illustrate the robustness of the model, we have added the analysis of the matching equilibrium hypothesis. Meanwhile, we adjusted the structure of the article by combining the previous comments. (line 386-392)

  1. What are the policy implications of this study?

Response: Thank you for your comment.

This study has some policy implications. It suggests that competition designers should be aware of the irrationality of government action and the heterogeneity among cities. As the public choice school emphasizes, government action often fails. But they also acknowledge that good institutional design can avoid this situation. From the results of the article, it is clear that the honor competition is a method for sustainable development of the industrial sector. (line 496-498)

However, we need to emphasize that this study has a unique political background. The public choice theory assumes that natural migration of population reflects competition between cities. But does this conclusion still hold if the household registration system is in place?

In China, there are many restrictions on the movement of people. This is why honor competition is necessary for the development of sustainable industrial sectors in China. In Japan, there are not many restrictions on the movement of persons. However, evidence from Japan shows that the growth of Tokyo has caused the decline of many Japanese cities.

Thus, we mention political effect at the end when we explain the limitations of our study. Specific policy recommendations must be integrated with the political background of the country. We aim to remind the competition's designers of this. Design honor competition is more likely to compensate for the lack of urban competition.

Reviewer 2 Report

Dear author,

I have reviewed your article titled "The Impact of City Ranking on Industry Shifting: An Empirical Study" and found it to be a comprehensive analysis of the impact of city honor competition on the shift from the manufacturing sector to the service sector in Beijing. The research question is clear, the methodology is well-designed, and the results are supported by robustness and placebo tests.

However, I believe that the article could benefit from more detailed discussions of the potential mechanisms behind the observed shift from manufacturing to services. It would be valuable to explore how exactly the competition motivates governments to create sustainable economic structures through ecological criteria and how this leads to a shift toward the service sector. Additionally, the potential negative consequences of such a shift, such as job losses or a decline in overall economic output, should be discussed.

Furthermore, it would be valuable to examine the generalizability of the findings to other cities in China or other countries. The study only considers the impact of city honor competition on the shift from manufacturing to services, while other factors such as technological change, globalization, and changes in consumer preferences could also be driving the shift.

Finally, the article could benefit from more detailed discussions of the limitations of the study. For example, the study only examines the impact of city honor competition on the sectoral structure of the economy in Beijing and does not consider other potential drivers of the shift. It would be valuable to acknowledge these limitations and their implications for the generalizability of the findings.

Generally, your article provides valuable insights into the impact of city honor competition on the sectoral structure of the economy in Beijing. However, further research is needed to fully understand the mechanisms behind this shift and to examine the generalizability of the findings to other contexts.

Thank you for your contribution to the field.

Author Response

Response Letter

We are thankful for the reviewers and editor’s comments. The manuscript is revised according to the reviewer’s advice. All changes were highlighted in RED in the manuscript.

Reviewer 2:

Dear author,

I have reviewed your article titled "The Impact of City Ranking on Industry Shifting: An Empirical Study" and found it to be a comprehensive analysis of the impact of city honor competition on the shift from the manufacturing sector to the service sector in Beijing. The research question is clear, the methodology is well-designed, and the results are supported by robustness and placebo tests.

However, I believe that the article could benefit from more detailed discussions of the potential mechanisms behind the observed shift from manufacturing to services. It would be valuable to explore how exactly the competition motivates governments to create sustainable economic structures through ecological criteria and how this leads to a shift toward the service sector. Additionally, the potential negative consequences of such a shift, such as job losses or a decline in overall economic output, should be discussed.

Furthermore, it would be valuable to examine the generalizability of the findings to other cities in China or other countries. The study only considers the impact of city honor competition on the shift from manufacturing to services, while other factors such as technological change, globalization, and changes in consumer preferences could also be driving the shift.

Finally, the article could benefit from more detailed discussions of the limitations of the study. For example, the study only examines the impact of city honor competition on the sectoral structure of the economy in Beijing and does not consider other potential drivers of the shift. It would be valuable to acknowledge these limitations and their implications for the generalizability of the findings.

Generally, your article provides valuable insights into the impact of city honor competition on the sectoral structure of the economy in Beijing. However, further research is needed to fully understand the mechanisms behind this shift and to examine the generalizability of the findings to other contexts.

Response: Thank you for your valuable comments.

The study is lack a more nuanced description of the incentives and negative effects of competition. We add more contents in Section 3.

For competitive incentives, according to the Liu’s research, it finds that 70.59% of the Mayors and 66.67% of the Municipal Party Secretaries who received the honor are promoted in 5 years. More than half of the senior officials are promoted before their next reassignment after being honored. And we added public choice theory to illustrate the motive of officials. Political incentives have prompted the government to set stricter environmental standards. (line 142-145)

On the other hand, the migration of residents is also a very important influence. Competition between cities can actually be considered as competition for talent. This means that innovation-intensive industrial sectors often require a livable urban environment. Just as Silicon Valley does. This is one of the reasons why government set stricter environmental standards. City honors are a signal to attract people to move to this city. (line 157-161)

And as for the negative impact of honor competition, these environmental costs are transferred to two groups, workers and consumers. This depends on the substitutability of the product. If the product is not substitutable, then consumers will be forced to accept the result of higher product prices. If the product has many substitutes, it is not a wise choice to raise the price. Dismissing workers to reduce costs is the main choice of the company. We tend to believe that both consumer and worker interests will be damaged. We add all these contents to our exposition. (line 187-192)

Regarding the study limitations, we strongly agree with your comments. Shifting in economic sectors can be influenced by many factors. Human capital, globalization, and technological progress can all be studied as control variables. That is why we give the range of effects of the urban honor race. We believe that with further research we believe we can get a more accurate effect. We will make a clearer statement about the limitations. (line 502-504)

Finally, on the generalizability of our study. This study has some policy implications. It suggests that competition designers should be aware of the irrationality of government action and the heterogeneity among cities. As the public choice school emphasizes, government action often fails. But they also acknowledge that good institutional design can avoid this situation. From the results of the article, it is clear that the honor competition is a method for sustainable development of the industrial sector. (line 496-498)

However, we must emphasize that this study has a unique political background. The public choice theory assumes that natural migration of population reflects competition between cities. But does this conclusion still hold if the household registration system is in place?

In China, there are many restrictions on the movement of people. This is why honor competition is necessary for the development of sustainable industrial sectors in China. In Japan, there are not many restrictions on the movement of persons. However, evidence from Japan shows that the growth of Tokyo has caused the decline of many other cities.

Thus, we mention political effect at the end when we explain the limitations of our study. Specific policy recommendations must be integrated with the political background of the country. We aim to remind the competition's designers of this. Design honor competition is more likely to compensate for the lack of urban competition.